# Estimating the Percentage of a Population Infected with SARS-CoV-2 Using the Number of Reported Deaths: A Policy Planning Tool

**DOI:** 10.3390/pathogens9100838

**Published:** 2020-10-13

**Authors:** Daniel R. Feikin, Marc-Alain Widdowson, Kim Mulholland

**Affiliations:** 1Independent Consultant, 1296 Geneva, Switzerland; 2Institute of Tropical Medicine, 2000 Antwerp, Belgium; mawiddowson@itg.be; 3Murdoch Children’s Research Institute, Royal Children’s Hospital, Flemington Road, Parkville, Melbourne 3051, Australia; kim.Mulholland@lshtm.ac.uk; 4London School of Hygiene and Tropical Medicine, London WC1E 7HT, UK

**Keywords:** SARS-CoV-2, coronavirus, Covid19, sero-prevalence, serology, low- and middle-income countries

## Abstract

The magnitude of future waves of Covid19 in a population will depend, in part, on the percentage of that population already infected, recovered, and presumably immune. Sero-epidemiological surveys can define the prevalence of SARS-CoV-2 antibodies in various populations. However, sero-surveys are resource-intensive and methodologically challenging, limiting widespread use. We propose a relatively simple method for calculating the percentage of a population infected, which depends on the number of reported Covid19 deaths, a figure usually more reliable and less dependent on variable testing practices than the total number of reported Covid19 cases, and the infection fatality rate, a figure that is relatively stable in similar populations. The method can be applied in different sized areas, such as states, districts, or cities. Such an approach can provide useful, real-time estimates of probable population immunity in settings unable to undertake multiple sero-surveys. This method is applicable to low- and lower-middle-income country (LMIC) settings where sero-survey data will likely be limited; however, better estimates of infection fatality rates and Covid19 death counts in LMICs are needed to improve the method’s accuracy. Information on the percentage of a population infected will help public health authorities in planning for future waves of Covid19, including where to most effectively deploy vaccines.

## 1. Introduction

Until SARS-CoV-2 vaccines are available, the magnitude of future waves of Covid19 in a given geographic area will depend, in part, on the percentage of that population already infected with SARS-CoV-2. If infection leads to immune protection, as current data suggest for at least the short term, the more people who have been infected with SARS-CoV-2, the fewer will be susceptible to become infected and severely ill in subsequent waves [1,2,3,4]. When the fraction of a community that is immune surpasses the so-called herd immunity threshold, the spread of the virus will slow and eventually stop. The exact immunity threshold is unknown, estimated to be 45–75% of the community being immune, and depends on local factors such as population density, host factors, contact patterns and control measures in place [5]. The true level of infection in a given community, however, is difficult to ascertain. The number of people diagnosed with Covid19 represents only a fraction of the true number of infections, and does so to an even greater degree in low- and lower-middle income countries (LMICs) where testing has been limited [6]. 

With the development of antibody tests, more sero-epidemiological surveys (sero-surveys) are being undertaken to define the prevalence of SARS-CoV-2 antibodies in various populations. However, sero-surveys are costly, consume staff time, require large populations, and are methodologically challenging to provide unbiased, representative data. Therefore, sero-surveys, especially repeated sero-surveys during the course of the pandemic, will likely be impractical to perform in most settings. To date, few sero-surveys have been undertaken in LMICs [7]. Public health authorities need additional accessible methods to understand the levels of infection and immunity in their communities. We propose the use of the number of reported Covid19 deaths in a defined population, a data point that is often available and more accurate than the number of Covid-19 cases, to make a relatively simple calculation that estimates the percentage of the population already infected. This approach can be applied at a national or subnational level. Knowing the percentage of a population that has been infected will help in planning for future waves of Covid19, including preparing the hospital system, procurement of supplies and medication, as well as prioritizing distribution of future vaccines. 

### 1.1. Why Covid19 Case Counts Are an Unreliable Indicator of the Percent of the Population Infected

Early in the pandemic, the availability of PCR test kits was limited in many countries. Therefore, testing was restricted, often based on the presence of risk factors or clinical severity. Insufficient testing capacity is still a major problem in many countries at this stage of the pandemic. In most places, many infected people with mild-to-moderate disease, estimated to be up to 80% of all infections, are not tested, either due to limited testing capacity or because they never present for testing [8]. Under-testing is even a greater problem in LMICs. In sub-Saharan Africa and South Asia, Covid19 testing rates are on average only 5–8% of testing rates in western European countries [6]. According to recent sero-surveys, in most populations, the true number of infections is many-fold higher than that indicated by routine testing [9,10,11,12].

### 1.2. Why Sero-Surveys Won’t Be Feasble in All Settings

Most people with SARS-CoV-2 infection, even with mild illness, produce antibodies within a few weeks of infection [4,13]. Therefore, well-executed sero-surveys are the best method to determine the percentage of a population that has been infected. However, sero-surveys have some limitations. Sero-surveys can yield biased results if not carefully undertaken [14,15]. This can occur through non-representative sampling of the total population (e.g., blood donors, residual samples from clinical labs), or self-selection among intended random samples (e.g., over-participation of those who suspect they had Covid19) [12,16]. Moreover, large, geographically-distributed samples are needed to obtain representative seroprevalence results for a population. In addition, the test performance of the serological assay can result in erroneous seroprevalence estimates if not corrected for, particularly in low prevalence populations when using serological tests with imperfect specificity [15]. Lastly, sero-surveys may not represent populations in specific settings with increased infection and/or death rates, such as care-homes or long-term care facilities (LTCFs) in high-income countries (HICs) and overcrowded slum communities in LMICs. For example, preliminary results from a sero-survey in Mumbai, India showed 58% of slum-dwellers versus 17% of the non-slum population had antibodies to SARS-CoV-2 [17]. Because of these considerations, as well as the resource limitations in LMICs, sero-surveys will likely only be undertaken in a limited number of LMIC settings.

### 1.3. Why Covid19 Death Counts Are Likely to Be More Accurate Than Case Counts

In contrast to people with mild-moderate Covid19 disease, severely ill persons are more likely to be tested for SARS-CoV-2. Most testing algorithms prioritize severely ill persons. Because the median time from symptom development to death is estimated to be 2–3 weeks, most people who die are likely to have presented to a hospital to be tested before dying, in settings with reasonable access to care and testing [18].

To test this hypothesis, we compared the association between SARS-CoV-2 testing rates and case and death rates among European countries that were past peak circulation in early June. (Figure 1) The rate of Covid19 cases reported was significantly associated with the testing rate (*p* = 0.001), while the death rate had no association, suggesting that total case notification rates are highly dependent on testing rates, while death notifications are less so. Because of very low testing rates in the majority of LMICs, the associations between testing rates and case/death rates are difficult to evaluate in these settings.

### 1.4. The Infection-Fatality Rate (IFR) Should Be Relatively Stable among Similar Populations

An important parameter is the infection-fatality rate (IFR), which is the number of total deaths per the number of total infections, or the proportion of all infected people who die. IFRs can be obtained through sero-surveys that determine population infection percentages or from models of infection and death prevalence [19,20]. The IFR, which is based on the number of total infections rather than only PCR-diagnosed cases, is less influenced by variability of testing between populations than the case-fatality rate (CFR). Sero-surveys done to date suggest that the rate of infection is similar across most age groups (except young children and perhaps very elderly), in contrast to the rate of symptomatic and especially severe disease, which increases with age [9,10,12,21]. IFRs from studies done to date in Europe, North America, and East Asia seem to be relatively consistent across different settings [19,22]. A meta-analysis of 25 IFRs found a summary IFR estimate of 0.68, with 95% confidence intervals 0.53–0.82 [19]. However, IFRs in some settings were outside this range due to methodological and real epidemiological factors, as discussed below. Moreover, no sero-surveys from LMICs were included in the meta-analysis.

## 2. Materials and Methods

### Calculation of the Percent of the Population Infected

The number of Covid19 deaths in a defined geographic area should be available from public health agencies. Usually, this includes both lab-confirmed and probable Covid19 deaths. The current death count will estimate the percent of a population infected several weeks earlier, an important factor if the number of cases is still increasing rapidly; however, the method can be applied at several different time points in a dynamic transmission setting. In places in which total Covid19 deaths might be under-reported, excess mortality could be used to estimate total Covid19 deaths, if reliable historical death data exist, to serve as upper bound of the estimate [23,24,25].

The other data point necessary for the calculation is the population residing in the study area. This should be available from local population or census departments. The denominator should represent the same population from which the number of Covid19 deaths was obtained. In rural areas of LMICs, this may be difficult to ascertain, so analysis could be limited to deaths occurring in a defined sub-population in which deaths are well-categorized. For the IFR, a summary estimate from multiple studies can be used. At the time of this writing, the IFR from the afore-mentioned meta-analysis will be used as the best available summary estimate to date—0.68 (95% CI, 0.53–0.82); however, as mentioned, this IFR is unlikely to be accurate for LMIC populations [19]. As more sero-surveys are done in different parts of the world, it will possible to generate summary IFR estimates for different strata of countries, categorized by factors likely to influence the probability of dying, such as age structure, prevalence of comorbidities, and access to critical care [26,27].

There are two steps to calculate the percent of the population infected with SARS-CoV2 from Covid19 deaths (Box 1). First, divide the number of Covid19 deaths by the summary IFR to get the number of SARS-CoV2 infections. Note that the IFR is usually reported as a percentage but should be converted to a proportion in this step. Moreover, the 95% confidence intervals of the IFR can be used to provide an uncertainty range. Lastly, divide the number of infections by the population denominator, multiplying by 100, to give the percentage of the population estimated to have had SARS-CoV-2 infection.

Box 1Calculation of the population infection incidence from the number of Covid19 deaths.**D—** Number of Covid19 deaths in a defined geographic area (from official reports).**IFR**—Infection Fatality Rate (as a proportion, from meta-analyses of IFRs from similar populations)**I**—Number of SARS-CoV-2 infections in defined geographic area (calculated)**Pop**—Current Population in defined geographic area (from census or statistical office)**PIP**—Percent infected in the population with SARS-CoV-2**Step 1**. D/IFR = I**Step 2.** (I/Pop)*100 = PIP

All data used in calculations were accessed from available public websites, as indicated. Calculations were performed in Microsoft Excel for Mac (version 16.16.24). Associations between independent variables were assessed using the linear regression function in Excel.

## 3. Results

Table 1 shows the results of applying this method in several countries with differing testing practices and Covid19 epidemiology.

### Validation of Method

A sero-survey done in Geneva in the first week of May revealed an age- and sex-adjusted sero-positivity of 10.8% (95% CI, 8.2–13.9) [12]. This is similar to the percent infected determined by the method proposed here (8.1%, 95% CI 6.7–10.4). A nationally representative sero-survey in Spain undertaken from April 27–May 11 showed a national seroprevalence of 5.0% (4.7–5.4), with a seroprevalence in Madrid of 11.3% (9.8–13) and Galicia of 2.1% (1.7–2.6) [10]. The percent infected from the national sero-survey was somewhat lower than we found using the number of reported deaths (8.8%), especially in Madrid. The summary IFR from the meta-analysis might have been too low for the Spanish population, which has an older age structure in general and had a disproportionate number of deaths among residents of LTCFs in the cities (approximately two-thirds); LTCF residents were excluded from the sero-survey [28]. The infected percentage estimated by the sero-survey and our method, however, were likely close enough in magnitude to make similar public health inferences (e.g., not close to herd immunity threshold).

Applying the death count method to Kenya yielded an infected percentage of the population of 0.046% in mid-June, when using the summary IFR of 0.68% (Table). In contrast, a recent national sero-survey among 15–64 year old blood donors in Kenya revealed an adjusted seroprevalence of 5.2% in May–June [21]. This percentage is over ten-times higher than that predicted by the death count method. Two factors can explain this discrepancy. First, Covid19 deaths could be undercounted in Kenya. This might be particularly the case among elderly people living in rural areas, where healthcare utilization is lower and Covid19 testing was likely less common than in urban centers. However, excess mortality estimates in Kenya, or other lower income countries, are not available to confirm this possibility [24,25]. Second, the summary IFR from the meta-analysis of 0.68% is certainly too high for the Kenyan population. Persons >60 years old account for only 5% of the total Kenyan population and 6% of Covid19 cases, yet 43% of Covid19 deaths [29,30]. Based on the Kenyan sero-survey using a seroprevalence of 5.2% would yield a crude IFR of 0.006%. Of note, the sero-survey excluded persons <15 and >65 years of age who make up almost half of the Kenyan population, and if as in other sero-surveys the sero-prevalence in these age groups is lower than that of the sero-surveyed population, the IFR for Kenya would be higher than 0.006%. Indeed, subsequent modelling work by Ojal et al. from the Kenya sero-survey estimated an age-adjusted IFR of 0.014–0.02% in the two largest cities [31]. Because no other African countries have published a representative national sero-survey, we provide a hypothetical example of how an “African IFR” might be used with the proposed method. If we apply an average IFR of 0.017% from the Ojal study to Ghana, and use the Covid19 death counts in Ghana on 21 May, 21 July and 21 September, we would estimate the infected percentage of the Ghanaian population as 0.58%, 2.9%, and 5.6% on 1 May, 1 July, 1 September, respectively, given approximately a three-week lag between infection and death [32].

## 4. Discussion

We provide a simple method based on Covid19 death counts for estimating the percent of a population infected with SARS-CoV-2, which can be applied at the level of discrete geographic areas with defined denominator populations (e.g., state, district, city). The percent of a population infected can inform estimates of the level of population immunity, which can guide planning for future SARS-CoV-2 waves and allocation of resources, including where to prioritize use of eventual vaccines. While this method should not replace well-executed sero-surveys, this method can provide infection percentages in places that will not perform sero-surveys, such as many LMICs. We call for more reliable counting of deaths and more sero-surveys in LMICs to be able to employ this method with greater validity in these settings.

The greater reliability of reported Covid19 deaths than cases, and the ability to back-calculate Covid19 infections from Covid19 deaths has been pointed out previously. The association between deaths and infections has been incorporated in models to calculate the rate of infection, as well as the impact of interventions [33,34,35]. Our method differs in that it relies on data from the IFR from a growing number of sero-surveys that directly collect data on antibody prevalence. The proposed method uses widely available data and can be done without modelling techniques. The calculations can also be done repeatedly using the latest death count data to adapt public health approaches during the course of the pandemic.

Despite its ease of use, the proposed method has several limitations. First, despite likely being more reliable than case counts, Covid-19 deaths can also be under-counted due to several reasons—limited testing availability, lack of clinical suspicion of Covid-19, and suboptimal access to health care and testing among pre-fatal cases. All of these issues are likely more widespread in LMICs [36,37]. In addition, due to political reasons, Covid19 deaths are not accurately reported in some countries. Several approaches might be taken to obtain more accurate death counts in LMICs. If deaths have likely been undercounted at a national level, there might be more circumscribed populations in the country where Covid19 deaths have been well-documented. For example, some countries have demographic surveillance systems where deaths are actively monitored [38]. In addition, defined catchment populations for sentinel hospitals already involved in surveillance for other diseases where hospital utilization is high might more reliably enumerate Covid19 deaths [39]. In countries or regions with vital statistics and death registration, excess mortality approaches that estimate the increase in all-cause mortality in certain age groups during the pandemic compared to prior years might be used [25]. Lastly, systematic surveys of defined populations to assess likely Covid19-related deaths could be undertaken. Assuming that these populations are similar in terms of age-structure and comorbidities with the rest of the country, they could provide estimates of infection percentages nationally.

A second, more significant, limitation of this method is the assumption that the IFR is a relatively stable construct across populations. While most IFRs calculated in upper-middle and high-income countries have fallen in a relatively narrow range, there is heterogeneity among the IFRs with some outliers [19]. Several papers have noted the poor quality of many sero-surveys, and the high risk of bias [14,19,26]. Undercounting of deaths in an area where a sero-survey was done can lead to an artificially low IFR, while exclusion of populations at high risk of death from sero-surveys, like LTCF residents, can artificially elevate IFRs. Future meta-analyses of sero-surveys should develop a set of criteria to exclude studies with clear methodological flaws and biases, as has been suggested [14]. Besides these methodological issues in calculating IFRs, much of the heterogeneity likely reflects true epidemiological factors. The key variable in differing IFRs is the age distribution of infected population. For example, the IFR aboard the Diamond Princess cruise ship that contained an older population (mean age 58 years) was estimated to be high (1.3%), while the IFR was 0 among U.S. Navy Service members aboard an aircraft carrier (median age 30 years) [40,41]. Recent modelling work has proposed summary estimates for IFRs for various age groups, which could be applied in an age-stratified approach to various countries according to their age stucture [42]. However, even age-specific IFRs might be differ in high-income and LMICs, particularly among the elderly [42,43]. Other factors that might account for this country variance in IFR are the presence of co-morbidities, the health system’s capacity to support critically ill patients, and differences in the percentage of the elderly living in LTCFs [27,42,44].

Importantly, most IFR estimates have been derived from data from high- and upper-middle income countries. Early evidence suggests that IFRs are likely lower in LMICs. We described how the IFR in Kenya based on a sero-survey is likely much lower than the 0.68% figure from the meta-analysis of IFRs. Sero-surveys in Malawi, Nigeria and India also have suggested much lower IFRs in these settings [36,45,46]. These differences in IFR are likely due to significantly younger and healthier populations in these countries [42]. In addition, possible environmental factors and/or partial immunity to SARS-CoV2 from previous exposure to coronaviruses might play a role in the lower mortality from Covid19 in LMICs [21]. More well-executed sero-surveys are needed from LMICs in Africa and Asia, allowing for calculation of summary IFR’s more appropriate for these parts of the world; more of these sero-surveys are being undertaken and are being supported by WHO as part of the Solidarity II and UNITY global serological studies collaboration and the Africa Centers for Disease Control and Prevention [47,48]. An alternative approach possible for LMICs is the use of one or a few well-executed sero-surveys in representative populations in a large country, like India or Indonesia, which can establish a relevant national IFR that can be applied serially throughout the country, or in neighboring countries with similar population structure.

Lastly, questions remain about the durability of antibodies to SARS-CoV-2. For SARS and MERS, antibodies started to wane 1–3 years after infection [2,49]. If antibodies to SARS-CoV-2 wane, the percent of the population having been infected would diverge from the percentage of the population with antibodies. Recent evidence suggests that antibodies to some serology test targets among asymptomatic and pauci-symptomatic Covid19 cases are no longer detectable within several months, while other studies using different antibody targets have observed longer-term durability [4,50,51]. The kinetics of the antibody response to various targets used in sero-surveys along the spectrum of Covid19 disease still needs to be better defined.

Although much is still unknown about how the Covid19 pandemic will evolve over the coming months and years, estimating the percent of the population infected can be a useful data point in predicting and planning the future response. We provide a method that we feel can be employed relatively easily using local data available in many LMICs to inform public health policy. We call for better enumeration of Covid19 deaths and better estimates of IFR for LMICs to make this method more applicable in these settings.

## Figures and Tables

**Figure 1 pathogens-09-00838-f001:**
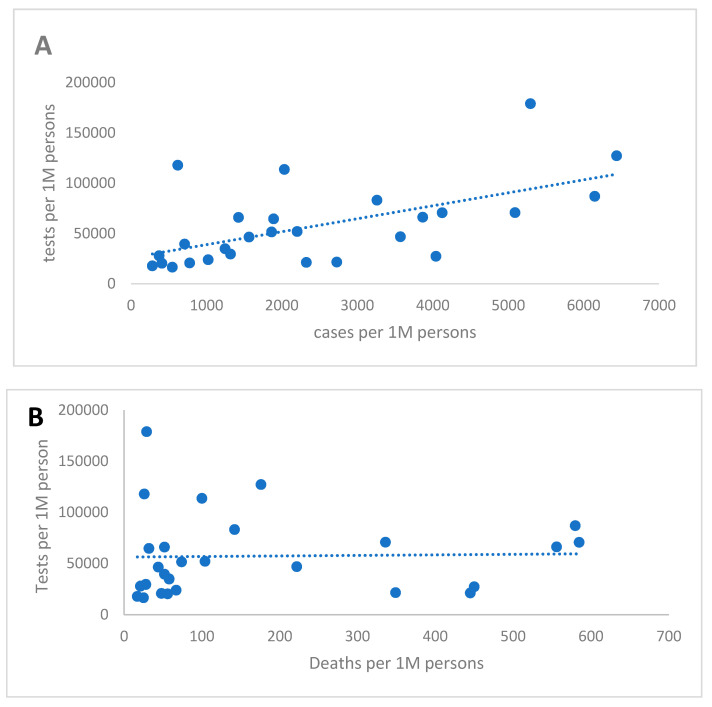
Testing rate of SARS-CoV2 by PCR test vs. case rate (**A**) and death rate (**B**), Europe, 3 June 2020 (Includes 27 European countries past peak of Covid19 case counts on 3 June 2020. Countries with rising or flat case counts were excluded as deaths lag behind cases by several weeks. Linear regression line shown as the dotted line; for case rates adjusted R-squared 0.32, *p* value = 0.001, and for death rates adjusted R-squared−0.039, *p* value = 0.9. Data from: https://www.worldometers.info/coronavirus/#countries and https://gisanddata.maps.arcgis.com/apps/opsdashboard/index.html).

**Table 1 pathogens-09-00838-t001:** Examples from several countries of the proposed method for calculating the percent of the population infected with SARS-CoV2 from the number of deaths, 4 June 2020.

1. Location	2. Number of Reported Covid19 Cases	3. Population ^a^ (‘Pop’ in Box)	4. Percentage of Population Reported as Covid19 Cases (Column 2/Column 3) ×100	5. Number Reported Covid19 Deaths (‘D’ in Box)	6. Calculated Percentage of Population Infected Using Proposed Method ^a^ (95% CI) (Calculated as ‘PIP’ in Box)	7. Ratio of Percentage Infected to Reported as Cases (95% CI) (Column 6/Column 4)
Switzerland ^b^	30,974	8,688,200	0.36%	1922	3.2% (2.7–4.2)	9 (8–12)
Zurich	3631	1,553,600	0.23%	130	1.2% (1.0–1.6)	5 (4–7)
Geneve	5158	509,100	1.0%	281	8.1% (6.7–10.4)	8 (7–10)
Spain ^c^	233037	46,660,000	0.50%	27940	8.8% (7.3–11.3)	18 (15–23)
Madrid	67049	6,685,471	1.0%	8931	19.6% (16.3–25.2)	20 (16–25)
Galicia	9077	2,698,764	0.34%	608	3.3% (2.8–4.3)	10 (8–13)
Kenya ^d^	8250	53,916,841	0.015%	167	0.046% (0.038–0.058%)	3 (2–4)

^a^ IFR used in calculation of summary estimate from meta-analysis 0.68% (95% CI 0.53–0.82%) [22]; ^b^ Swiss cases and deaths of 4 June 2020, https://ddrobotec.maps.arcgis.com/apps/opsdashboard/index.html#/5ed2e108dbab4235a7318d1cfe147e7a. Population from 2020 https://www.bfs.admin.ch/bfs/en/home/statistics/population/population-projections/cantonal-projections.html; ^c^ Data on Covid19 cases in Spain on June 4 and population projections for 2019 from https://www.zoho.com/covid/spain/ and https://www.statista.com/statistics/445549/population-of-spain-by-autonomous-community/; ^d^ Data for Kenya taken on 7 July from https://www.worldometers.info/coronavirus/#countries. 7 July was chosen because it was approximately 3 weeks after the last day of the serosurvey to account for a lag in time from infection to death [21].

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
