# Peer review of "Estimating the Percentage of a Population Infected with SARS-CoV-2 Using the Number of Reported Deaths: A Policy Planning Tool"

_pathogens, 2020, doi:10.3390/pathogens9100838_

Round 1
Reviewer 1 Report
Very well written paper on a very relevant subject. Innovative ”simple” approach to estimate the percentage of population infected with SARS-CoV-2.
I have no major comments
However for LMIC, as the authors mention in their paper, there are many problems to use this “simple” method.
Most importantly, reliable COVID-19 mortality data are generally not available.
It would be interesting to know whether the authors have suggestions how to obtain more reliable COVID-19 mortality data.
Could it be that one of the reasons of the low reported COVID-19 mortality in Africa may be that older people with COVID-19 die at home and that because of COVID-19 related stigma their death is not reported?
The authors mention that the IFR in Kenia most likely is much lower than in Europe but this seems to be the case in most sub-Saharan African countries. It could be interesting to highlight in this paper the different evolution of the COVID-19 epidemic in sub-Saharan Africa compared with other LMIC. Such a different epidemic requires also a different approach to control it.
The question can also be asked how important it is for public health decision makers to know the exact COVID-19 prevalence in African populations if infection is associated with very low mortality. Strengthening surveillance for COVID-19 related severe disease/mortality seems to me more important than sero-surveys. This also means that control efforts should be targeted at protecting those at risk for severe disease and not to stop transmission with lockdown measures. I leave it to the authors to consider to include some of these reflection in their paper.
Author Response
Reviewer 1.
Very well written paper on a very relevant subject. Innovative “simple” approach to estimate the percentage of population infected with SARS-CoV-2.
I have no major comments.
However, for LMIC, as the authors mention in their paper, there are many problems to use this “simple” method.
Most importantly, reliable COVID-19 mortality data are generally not available. It would be interesting to know whether the authors have suggestions how to obtain more reliable COVID-19 mortality data.
Authors’ Response: We now offer further suggestions about how to obtain better estimates of death counts in LMICS. This is on lines 314-325.
Could it be that one of the reasons of the low reported COVID-19 mortality in Africa may be that older people with COVID-19 die at home and that because of COVID-19 related stigma their death is not reported?
Authors’ Response: While this is possible, enough time has passed now (> 6 months) since the virus has been present in Africa that there should be reports of older people having died in African villages, or excess community deaths captured by demographic surveillance. To date, there have not been reports of this in any African country, to our knowledge.
The authors mention that the IFR in Kenia most likely is much lower than in Europe but this seems to be the case in most sub-Saharan African countries. It could be interesting to highlight in this paper the different evolution of the COVID-19 epidemic in sub-Saharan Africa compared with other LMIC. Such a different epidemic requires also a different approach to control it.
AUTHORS’ RESPONSE: We agree that the IFR is likely equally low in other African countries with similar demographics. We state this on lines 343-344 and have added another example from Niger state Nigeria – all these studies show relatively high seropositivity with low death counts leading to a low IFR. A discussion about differences in the epidemiology of COVID-19 in Africa and how that translates into differences in approaches to control is certainly interesting and important, but beyond the scope of this paper.
The question can also be asked how important it is for public health decision makers to know the exact COVID-19 prevalence in African populations if infection is associated with very low mortality. Strengthening surveillance for COVID-19 related severe disease/mortality seems to me more important than sero-surveys. This also means that control efforts should be targeted at protecting those at risk for severe disease and not to stop transmission with lockdown measures. I leave it to the authors to consider to include some of these reflection in their paper.
AUTHORS’ RESPONSE: We agree that the emerging epidemiology in Africa suggests focusing on protecting those at highest risk. We still believe there is a role for sero-surveys in the African setting for several reasons:
- There still are only a handful of sero-surveys from Africa and we need to have more robust data before we can confirm the early epidemiological inferences.
- Knowledge of the percent of the population infected can help make decisions about approaches to prevention. For example, it can help target certain groups for vaccination or in extreme cases when infection prevalence is very high, lead public health authorities to put less emphasis on the need for extensive vaccination.
No changes have been made in the paper related to this comment.
Reviewer 2 Report
This paper examines the use of reported COVID-19 deaths as a proxy for computing the number of people that have been infected, as a potential substitute for sero-surveys. The paper is well written and comprehensively discusses the applicability and limitations of the proposed method, and the authors were clear and realistic about when their approach can be used. A way to easily estimate the percentage of a population that had been infected would be valuable in the near future, particularly in LMICs, and this paper seeks to answer a timely and important question.
The proposed approach seems to boil down to - if both the infection-fatality rate in a setting and the number of deaths are known, they can be divided to estimate the number of infections. Given that this follows more or less directly from the definition of the IFR I think the value-add of a study like this one would be less in the novelty of the method, and more in validation of the method. In particular, it would be valuable for policy-makers or other researchers to have a clear idea of whether this calculation would yield sufficiently accurate results in their setting given their circumstances, so that they could use estimates from this method in subsequent analyses.
In that regard, I found the analyses in this paper somewhat limited. The study examined 7 settings as shown in Table 1 (nb. in the text under the table, item (d) suggests that maybe there is a row missing from the table?). The Spanish settings were overestimated compared to the sero-prevalance study in that country, and it was argued that this was because new cases were still occurring, the IFR may have been too low, and/or the sero-prevalance study excluded LTCFs with high mortality. In Kenya, the method considerably underestimated the proportion infected, which was argued may be the result of under-reporting of COVID-19 deaths, and the IFR being too high, perhaps due to the population sampled for the sero-survey.
This leaves one wondering about the extent to which the approach could be trusted if sero-prevalence data were not available, which is the main appeal of this method in the first place. The discrepancies in Kenya highlight the importance of having IFR estimates from similar settings, which are as yet not available. The discrepancy in Spain is rather more difficult because European settings were included in the meta-analysis. The cited meta-analysis (which does not appear to have been peer-reviewed yet) notes that there is high heterogeneity in the IFR estimates, perhaps because the death rates depend on government responses, as well as the particular subset of people that have been infected in each country. This does not seem to bode well in the translation of IFR from one setting to another particularly given that there is a high degree of variability in government responses even for countries that may be otherwise comparable. Also, the suggestion that results are more accurate several weeks past the peak wave of cases could be quite limiting since there would likely be interest in estimating the size of the immune population even while cases are growing or steady. To that end, I think it would be useful to have a quantitative analysis of the extent to which growing or steady case counts bias the estimate, to give an indication of how this method may perform in settings that have not yet peaked.
Overall I think this work may be premature in that there would have been significantly more value if there were at least a few other reliable IFR estimates from LMICs available for comparison, as well as larger, more reliable sero-prevalance estimates that could serve as a ground truth for validating the proposed approach. Otherwise, if the validity of the sero-prevalance studies used for comparison are themselves are in question, it's difficult to tell whether the proposed method is yielding meaningful results that can be trusted for policy planning and decision making purposes.
Author Response
Reviewer 2.
This paper examines the use of reported COVID-19 deaths as a proxy for computing the number of people that have been infected, as a potential substitute for sero-surveys. The paper is well written and comprehensively discusses the applicability and limitations of the proposed method, and the authors were clear and realistic about when their approach can be used. A way to easily estimate the percentage of a population that had been infected would be valuable in the near future, particularly in LMICs, and this paper seeks to answer a timely and important question.
The proposed approach seems to boil down to - if both the infection-fatality rate in a setting and the number of deaths is known, they can be divided to estimate the number of infections. Given that this follows more or less directly from the definition of the IFR I think the value-add of a study like this one would be less in the novelty of the method, and more in validation of the method. In particular, it would be valuable for policy-makers or other researchers to have a clear idea of whether this calculation would yield sufficiently accurate results in their setting given their circumstances, so that they could use estimates from this method in subsequent analyses.
AUTHORS’ RESPONSE: Yes, the method is a simple mathematical rearrangement of the IFR. Its simplicity is part of its utility. We believe, however, that it is not immediately obvious to most people that by using the death count, you can work your way back to the infected percentage of the population. The paper makes that point explicit in several places – for example, lines 16-18 and 53-55, and 257-258.
In that regard, I found the analyses in this paper somewhat limited. The study examined 7 settings as shown in Table 1 (n.b. in the text under the table, item (d) suggests that maybe there is a row missing from the table?). The Spanish settings were overestimated compared to the sero-prevalence study in that country, and it was argued that this was because new cases were still occurring, the IFR may have been too low, and/or the sero-prevalence study excluded LTCFs with high mortality. In Kenya, the method considerably underestimated the proportion infected, which was argued may be the result of under-reporting of COVID-19 deaths, and the IFR being too high, perhaps due to the population sampled for the sero-survey.
AUTHORS’ RESPONSE: First, thank you for pointing out the error in the table footnotes. It has been removed. We have modified the interpretation of the Spanish study. Subsequent data revealed that the first wave had already peaked by the time of the first Spanish sero-survey – few further cases or deaths occurred after that and the sero-prevalence stayed same in rounds 2 and 3. So we have removed the explanation of lagged cases to explain the difference. The main reason seems to be that the IFR used was too low for Spain that has an older age structure and many LTCF residents. Moreover, we have readjusted the “expectations” of this method. It is not expected to get the same sero-prevalence as sero-surveys but, be close enough from a public health action perspective. Therefore, we have emphasized that the differences in Spain are only a few percentage points (5% vs. 8.8%) and would lead to similar public health interpretations and actions. See lines 231-233.
This leaves one wondering about the extent to which the approach could be trusted if sero-prevalence data were not available, which is the main appeal of this method in the first place. The discrepancies in Kenya highlight the importance of having IFR estimates from similar settings, which are as yet not available. The discrepancy in Spain is rather more difficult because European settings were included in the meta-analysis. The cited meta-analysis (which does not appear to have been peer-reviewed yet) notes that there is high heterogeneity in the IFR estimates, perhaps because the death rates depend on government responses, as well as the particular subset of people that have been infected in each country. This does not seem to bode well in the translation of IFR from one setting to another particularly given that there is a high degree of variability in government responses even for countries that may be otherwise comparable. Also, the suggestion that results are more accurate several weeks past the peak wave of cases could be quite limiting since there would likely be interest in estimating the size of the immune population even while cases are growing or steady. To that end, I think it would be useful to have a quantitative analysis of the extent to which growing or steady case counts bias the estimate, to give an indication of how this method may perform in settings that have not yet peaked.
AUTHORS’ RESPONSE: We also would like to check the method’s validity with other LMICS. The problem is that there are no other representative sero-surveys besides the Kenya one. We provide a hypothetical example of applying this method using the IFR derived from the Kenya study to Ghana -- see lines 249-255. We do this at three points in time over the course of the outbreak – May 1, July 1 and September 1. Without sero-survey data in Ghana to compare these estimates with, we cannot know if the estimates of infection percentage are correct. However, they do look reasonable and provide an infected percentage of 5.6% on September 1, which is not so different from the 5.2% found in the Kenya sero-survey done in early June. We chose Ghana as an example because it is about the same size as Kenya, has similar age structure and GDP, and likely has more reliable death counts than other African countries.
Overall I think this work may be premature in that there would have been significantly more value if there were at least a few other reliable IFR estimates from LMICs available for comparison, as well as larger, more reliable sero-prevalance estimates that could serve as a ground truth for validating the proposed approach. Otherwise, if the validity of the sero-prevalance studies used for comparison are themselves are in question, it's difficult to tell whether the proposed method is yielding meaningful results that can be trusted for policy planning and decision-making purposes.
AUTHORS’ RESPONSE: we agree with these points. As pointed out the Kenya study reveals that the IFR from the meta-analysis will not apply to LMICS. We now more strongly call for more sero-surveys in LMICS to establish an IFR more accurate for LMICS and better death counts before this method is used in LMICS – see lines 23-24, 263-265, 412-414. We also propose a new approach besides a LMIC meta-analysis of IFRs, whereby one country can do one or a few representative sero-survey to get the IFR for their own use or those of neighboring countries – see lines 351-354. We believe now is the time to call for better data in LMICS to be able to use this method going forward.
Other changes to note for the editor.
- Due to addition of text as requested by reviewers, we have deleted some text to keep within word limit. As such, we have tried to focus more on the LMIC perspective, as we believe that this method is most applicable in LMIC settings and as LMICS are the emphasis of this special edition of pathogens. For example, in the results section just after the table, we have removed a paragraph that compares Switzerland, Luxembourg and Spain, as this is less relevant to an LMIC setting.
- We have added a few references that have been published since the submitted version of this manuscript. We also deleted a few sentences that might have resulted in deleting some references. We did not update the references section using endnote as it seems that this has already been formatted according to journal format. We include new references as comments and kindly ask that you add them using the journal formatting and remove references that are no longer cited in the text.
Round 2
Reviewer 2 Report
The authors revisions have addressed all my concerns.